# Development and Validation of the Tuberculosis Risk Score for Smokers (TBRSS)

**DOI:** 10.3390/ijerph19126959

**Published:** 2022-06-07

**Authors:** Tengku Noor Farhana Tengku Khalid, Wan Mohd Zahiruddin Wan Mohammad, Nik Rosmawati Nik Husain, Razan Ab Samat

**Affiliations:** 1Department of Community Medicine, School of Medical Sciences, Universiti Sains Malaysia, Kubang Kerian 16150, Kelantan, Malaysia; drtnfarhana@student.usm.my (T.N.F.T.K.); rosmawati@usm.my (N.R.N.H.); 2Bachok District Health Office, Bachok 16300, Kelantan, Malaysia; razan.absamat@moh.gov.my

**Keywords:** tuberculosis, smokers, screening, risk score, content validity, face validity, TBSR, outcomes

## Abstract

Tuberculosis (TB) remains a significant public health issue worldwide. However, the effectiveness of TB screening programmes among smokers is still questionable. There is a need for a simple, reliable, and validated screening system for this risk population. This study aimed to develop and validate the tuberculosis risk score for smokers (TBRSS) in Kelantan, Malaysia. A case–control study was conducted on 159 patients (smokers with and without TB) between January and July 2020. Simple and multiple logistic regressions were applied to determine the variables to be included in the risk score. The cut-off points to determine a score indicating low or high risk for TB disease were obtained based on the receiver operating characteristics curve. Content validation was carried out through interviews with eight experts to measure each variable′s relevancy. The face validation was conducted among 20 health clinic staff. Seven variables were selected for inclusion in the risk score. The chosen cut-off point was 16 (out of 43), with 91% and 78% sensitivity and specificity, respectively. The scale-level content validity index was 0.83, while the face validity index scores for each element ranged between 0.85 and 1.00. The TBRSS can be considered a validated screening tool for use in screening TB disease risk among smokers, which potentially may lead to an increased detection of TB disease in the community.

## 1. Introduction

Tuberculosis (TB) has remained one of the leading public health concerns among infectious diseases worldwide. Despite the number of cases decreasing annually, the rate of decline is still slower than that targeted by the World Health Organization (WHO) [1]. The Sustainable Development Goals (SDG) set a target for TB cases to decline 4–5% annually; however, globally, the rate of decline achieved has been around 2% [2]. One of the causes for the decline is the low identification rate of tuberculosis cases, particularly in low and middle-income nations. The WHO reported that around 3.6 million people worldwide were missed and were not treated accordingly for TB-related diseases [3,4]. To increase the case detection rate for TB diseases, the WHO has advocated for systematic screening that targets the high-risk group for TB diseases [5].

Tobacco use has long been under suspicion for associations with the TB epidemic. Growing epidemiological and laboratory evidence indicate that cigarette smoking, the most common form of tobacco use worldwide, is a risk factor for development and death from TB [6]. Smokers are considered one of the high-risk groups for screening. The mechanism for how smoking may increase the risk for TB disease among the smoker population is still not well-understood, but it is likely to be closely related to how smoking alters the physiological function of lung tissue [7,8].

The estimate for TB incidence rate in Malaysia was 92 cases per 100,000 population, and the TB mortality rate was estimated at 4 cases per 100,000 population per year in 2019 [9]. To date, the smoker population in Malaysia is screened for TB when they visit clinics or participate in the quit smoking clinic program held in health clinics [4,10]. This is considered opportunistic screening for the smoker population. According to this screening programme, the majority of the smoker population has not been screened for TB disease. Only 10% of the smoker population has visited a health clinic [11]. The screening was undertaken solely based on the patient being an active smoker. It did not consider the exposure in the case of passive smokers or history of smoking. Furthermore, no preliminary screening was carried out before the smokers were asked to provide sputum or undergo a chest X-ray [4,8]. Having sputum tested for acid-fast bacilli or a chest X-ray is part of the diagnostic test for TB disease [12].

Even though the WHO has been promoting chest X-ray examination (CXR) as a useful tool that can be placed early in screening and triaging algorithms for systematic screening for active TB, there are still limitations, given that CXR has low specificity, significant interobserver variation, it is costly and there is poor access to high-quality radiography equipment and expert interpretation [13]. These are the additional barriers for promoting CXR as a large-scale programmatic, especially as a low cost and effective screening tool for smokers at the primary level of healthcare.

The necessity of developing a new screening tool for smokers with TB in the community can be explained by a few justifications. Firstly, the current national TB screening program in Malaysia for the smoker population only targets smokers who attended the quit smoking programs voluntarily at the health clinics. However, only a small proportion of smokers attended these clinics within one year. Secondly, without an improved and effective screening tool for TB among smokers in the community, healthcare practitioners may only see them at a later stage with complications due to TB. This will cause delays in TB diagnosis and treatment. Thirdly, there is currently no systematic approach to screening TB among smokers, except for chest X-ray examination. This warranted the need to approach and screen the smokers in the community other than the smokers who visited the health clinics.

Several risk scores or scoring systems have been developed and validated to identify active TB cases in the population. The scoring systems developed have been targeted to children, adolescents, HIV clinic attendees, and contacts of TB patients [14,15,16,17,18,19]. The scoring systems have used readily available parameters or variables, such as sociodemographic factors or standard clinical parameters, to determine the probability or the risk of patients having TB disease. However, smoking status has not been considered a factor in any existing scoring or screening. A study from Canada did initially consider smoking status as an important variable to be included. However, it was dropped when the analysis showed the factor was not significantly associated with TB disease [19].

Therefore, a more accessible, cost-effective, and convenient screening test is needed for the smoker population. The risk score will support the diagnosis of TB disease in this group and increase their awareness of their risk of having TB disease. This study aimed to develop and validate a risk score for TB disease in the smoker population called the tuberculosis risk score for smokers (TBRSS) in Bachok, Kelantan, Malaysia, in 2020.

## 2. Materials and Methods

### 2.1. Case–Control Study

A case–control study was conducted among smokers who visited health clinics in Bachok, Kelantan, from January until July 2020. A list of patients was obtained from the TB registry TBIS101A of the Bachok District Health Office, and sputum examination records (TBIS102A) were gathered from the district health clinics. TBIS101A is a proforma that is filled up by the clinicians who have diagnosed new TB patients, consisting of variables of patients’ treatment, biodata, medical history and health status, while TBIS102A is the record of sputum investigation carried out at the laboratory in health clinics. Simple random sampling was applied to select the respondents to be interviewed from those lists.

The cases were selected from among patients registered in TBIS101A with a history of either active or passive smoking, or being an ex-smoker, and aged 18 years and above during the recruitment period. An active smoker was defined as a person who was still smoking any cigarette within the last six months, while an ex-smoker can be defined as a person who had not been smoking at all for the past six months. A passive smoker was a person who was exposed to smoke because of a smoker present among the family members within the last six months. The controls were selected among patients aged 18 years and above who were registered in the health clinics and recorded in TBIS102A, with a known history of being an active, passive, or ex-smoker, with recorded negative results for TB screening. The patients selected for controls were not registered in TBIS101A. Similar to the cases group, records among the controls with incomplete contact numbers or addresses and patients who were unable to understand and give appropriate answers to the interviewer were excluded.

The sample size was calculated using Power and Sample Size Calculations software version 3.1.2. The proportion of patients with exposure not diagnosed with TB disease (*P*_0_) was 0.59, and the expected proportion of patients with exposure diagnosed with TB disease (*P*_1_) was 0.82 [15]. The significance level (*α*) was 0.05 with the power (1 − *β*) at 0.8. The cases to controls (m) ratio were set to 1:2. Thus, the total estimated sample size needed, considering the response rate, was 168.

The cases and controls selected were interviewed via telephone call or face-to-face interview. Verbal or written consent was obtained before the interviewer began to ask the questions. The patients were interviewed using a proforma, which consisted of 45 items under 5 domains. The patients’ records in the health clinic and the TBIS101A were also reviewed to ensure information validity. All data were collected in a Microsoft Excel document and analysed using R software version 3.1.6. Simple and multiple logistic regressions were applied to obtain the factors and scores needed to form the TBRSS.

### 2.2. Determination of Cut-Off Points

Next, the cut-off points to differentiate between those with a high risk of TB disease and those with a low risk of TB disease were determined using the sensitivity–specificity curve. The highest and the leftmost point on the curve were chosen to differentiate between high risk and low risk of TB disease. For a screening test, the sensitivity was targeted to be higher than 80%, and the expected false rate should be lower.

### 2.3. Content Validity and Face Validity

To ensure the validity of the TBRSS, content validation and face validation studies were conducted. Eight experts were selected for the content validation study, including a respiratory physician, family medicine specialists, public health physicians, and medical officers in charge of TB patients. They were asked to score self-directed proformas, consisting of the factors chosen from the development of the TBRSS. Each expert was asked to score each factor according to a Likert scale of ‘1’ to ‘4’, where ‘1’ indicated that the factor was not relevant at all to the risk score and ‘4’ indicated that the factor was highly relevant to the risk score. A response of ‘1’ or ‘2’ gave a score of 0 to the factor’s total score, while a response of ‘3’ or ‘4’ gave a score of 1 to the factor’s total score. The factor content validity index (F-CVI) was calculated by dividing the factor’s total score by the number of experts who responded. The scale-level content validity index averaging method (S-CVI/Ave) was calculated by dividing the summation of all the F-CVI scores by the number of experts who responded. Both the targeted F-CVI and S-CVI/Ave were at least 0.83 [20].

The face validation study selected 20 participants among the healthcare providers working in the health clinics in Bachok, Kelantan. The recruitment involved all levels of staff in the health clinics, including medical officers, assistant medical officers, staff nurses, community nurses, medical laboratory technicians, and healthcare assistants. They were all asked to fill in a self-directed proforma. The participants were asked to score the TBRSS according to the following five elements: (i) understanding of the instructions given, (ii) understanding of the sentences in the risk score, (iii) ease of scoring, (iv) type and size of the font used, and (v) the appropriateness of the arrangement on the TBRSS. Each participant was asked to mark each element according to a Likert scale of ‘1’ to ‘4’. A score of 1 was given for every mark of ‘3’ or ‘4’, and a 0 was given for every mark of ‘1’ or ‘2’ for each element. The total score for each component was calculated by adding all the scores from all participants. The face validity index (FVI) was calculated by dividing the total score by the number of participants who responded to each element. There was no cut-off point for the FVI value; however, the large value indicated a better-validated risk score [21].

## 3. Results

Table 1 compares the characteristics of patients based on the case–control study. The total response rate for the case–control study used to develop the risk score was 159 (94.6%). The majority were male (59.1%), non-diabetic (72.3%), HIV-negative (98.7%), and had no other immunosuppression (86.2%). There were a mixed number of active smokers (35.2%), ex-smokers (21.4%), and passive smokers (43.4%). The duration of exposure to smoke was longer among the cases group (mean (SD) = 23.9 (16.47) years) compared to the control group (mean (SD) = 18.4 (12.84) years). The respondents were also immunized with BCG (bacille Calmette–Guerin) vaccine (93.1%), had never been exposed to TB index patients (66.7%), and had no previous history of TB disease (93.7%).

According to their history of TB symptoms, most of the respondents had a history of cough (86.2%), no night sweats (85.5%), no chest pain (91.8%), and no history of hemoptysis (86.8%). The group of cases also had a longer duration of symptoms before the diagnosis of TB disease (median (IQR) = 30 (76) days)) compared to the group of controls (median (IQR) = 7 (11) days).

The simple logistic regression showed that ten variables were significantly associated with TB disease among the smoker population in Bachok (Table 2). All ten variables and cough were included in the multiple logistic regression analysis due to clinical importance. The coefficient values of each variable were obtained. Only seven variables were selected to be included in the risk score, which were as follows: smoking status, duration of exposure to smoke, a previous history of TB disease, cough, night sweats, significant weight loss, and duration of TB symptoms. Four variables were not included in the risk score. The analysis showed that the variables were not significantly associated with TB disease and the beta (*β*) coefficient value obtained for the analysis was not found in other studies. The team members also agreed to remove those four variables from the risk score to increase the efficiency of the risk score for the population.

To determine the score for each level of the variables, the *β*-coefficients of each variable (except smoking status) were divided by the lowest *β*-coefficient obtained among the variables (cough: *β*-coefficient = 0.267). The values were then rounded up to the nearest integer before adding one to give the lowest score at least one (1). Table 3 shows the final scoring allocated for each of the variables. Based on the sensitivity–specificity curve analysis, the cut-off point chosen for the risk score was 16, where the sensitivity value was 91%, the specificity value was 78%, and the false-risk result was 8.9% (Figure 1).

Eight experts (80.0%) responded to the content validation study, with a median (IQR) duration of experience of sixteen (14) years among them. The range of F-CVI values for all variables was between 0.88 and 1.00. The S-CVI/Ave value was 0.83 (Table 4a). All participants recruited responded to the face validation study. They included all levels of staff working in the health clinics in the Bachok District. The mean (SD) for the duration of experience working in the health clinics among the participants was 12.8 (6.58) years. The total number of participants for the face validation study was 20 (100.0%). Most participants gave higher scores for the four elements tested in the face validation study (Table 4b). Three participants gave a lower score for the size of the font used in the TBRSS. They preferred the size to be larger. Overall, the total FVI values for all elements were between 0.85 and 1.00.

## 4. Discussion

The variables included in the risk score were based on the simple logistic regression analysis. In the multiple logistic regression analysis, only four variables were significantly associated with TB disease among the smoker population in Bachok, Kelantan. Having fewer variables for the screening process is more reliable, as it may lead to less confusion and less time to conduct the screening process. However, it may also lead to false negatives. Patients at higher risk for TB disease may be considered low risk for TB disease when some factors are not considered [22]. For example, the duration of exposure to smoke is considered an essential factor when assessing the smoker population.

Smoking status was significantly associated with TB disease after adjusting for other variables. The different smoking status levels were essential to determine whether the patient had a high risk or low risk for TB disease. Being an active smoker or an ex-smoker is associated with a higher percentage of patients with severe lung disease, lung cavitation, and positive sputum culture for TB organisms. This could be an effect of tobacco smoke on lung ciliary function in response to bacteria [8,9]. Comparing active smokers and ex-smokers, studies have shown that an ex-smoker has a higher risk of TB disease than an active smoker [23,24]. One of the explanations for this finding could be due to poor health-seeking behaviour among the smoker population. In Malaysia, only about 10% of the smoker population visit a health clinic for any reason within a year-long period [8,25,26].

The cough symptom was included in the multiple logistic regression analysis, even though it was not significantly associated with TB disease in the simple logistic regression. This is because the symptom was deemed important to be included in the risk score as part of the screening for TB disease. The TB microorganism is mainly transmitted via respiratory symptoms, such as cough or sneezing [27]. The diagnosis of TB disease, particularly in patients with pulmonary TB disease, was made mainly using sputum smear microscopy.

Both the presence of significant weight loss and having night sweats were considered constitutional symptoms [28]. As TB is considered a chronic disease, both symptoms are important in diagnosing patients with TB disease. It was found that TB microorganisms promote the immune response responsible for tissue degradation and enhance the secretion of the protein responsible for the inhibition of appetite, causing patients to experience night sweats and weight loss. Exposure to smoke from cigarettes further increases tissue damage, reduces appetite, and causes a more significant immune response, causing night sweats [29,30,31].

Another variable included in the risk score was the duration of symptoms. Patients who had a longer duration of TB symptoms were more likely to have more positive smear in their sputum. Thus, they were more likely to be diagnosed with TB disease [31]. Our findings showed that those with TB disease had a longer duration of symptoms than those for whom TB disease was ruled out. Patients with a shorter duration of symptoms tend to seek treatment later [25]. However, prolonged treatment delay may further increase tissue damage, especially in the lungs [28,32].

When assessing smokers, it is crucial to consider the severity of the smoking behaviour, regardless of whether they are a current smoker or an ex-smoker. Unfortunately, it is difficult to quantify the severity of exposure to smoke among passive smokers. For the TBRSS, the variable was also included as part of the assessment. The higher the number of years of smoking the smoker had engaged in, the higher the risk of being infected with the TB organism and contracting TB disease. Prolonged exposure to tobacco smoke causes more damage to the lung ciliary function, which may be irreversible [7,9].

Similarly, those who had a previous TB history experienced lung tissue damage due to infection with the TB microorganism, thus causing the patient to be easily re-infected and manifest TB disease. Furthermore, the previous infection with the TB microorganism may have caused an increase in the innate immune response in the body, causing increased susceptibility to contracting TB disease in the future [33].

Many different types of risk scores have been developed to screen or diagnose TB disease in the population. In Brazil, the symptoms of cough, night sweats, significant weight loss, and duration of symptoms were included as variables in the risk score for diagnosing children with TB disease [17]. Hanifa et al. [14] also included cough, weight loss, and night sweats as part of their scoring system, to prioritize TB investigation among HIV clinic attendees in South Africa. Their study also showed that the duration of symptoms was significantly associated with TB disease among the HIV clinic attendees. To predict the risk for TB disease among contacts in Peru, Saunders et al. [18] created a score that includes the history of previous TB as a variable. Based on these previous findings, it is essential to include seven variables in the risk score as a screening tool for the smoker population.

Any instrument, screening tool, or questionnaire must be validated. Validating the TBRSS ensured that the tool will detect smokers with a high risk for TB disease quickly and efficiently for the patient [20,21]. The content validation study for the TBRSS focused on the relevancy of the chosen variables to be asked during the screening process to determine the risk for TB disease among the smoker population. Eight experts responded, which was sufficient to rate the variables [34]. Based on the result in this study, the F-CVI and S-CVI/Ave values were greater than 0.83. Many studies agree that an S-CVI/Ave value of 0.83 is highly acceptable [35,36,37].

The face validity study aimed to ensure that the screening tool is easily usable and understandable by the user who conducts the screening process. In this study, the FVI values for all items were more than 0.8. There was no cut-off point for the FVI value. However, a higher score is more favourable. Thus, the value of more than 0.8 indicated an acceptable face validity [21,38]. All respondents agreed that the instructions given on the TBRSS were easily understood, the sentences were understandable, and it was easy to mark the patient’s score on the TBRSS. The arrangement of the TBRSS was appropriate and made it easier and more comprehensible to use. Even so, three respondents gave a mark of ‘3′ for the type of font and size used on the TBRSS. They commented specifically on the size of the font used on the TBRSS. All three of them were more than 40 years old, with a mean age of 45.7 years. Even though they could read the sentences on the TBRSS, they were concerned that more senior users might have a problem reading the words and may misinterpret the risk score, especially those who have a visual impairment [39]. They suggested a bigger font should be used, if possible, to aid the use of the TBRSS. Nevertheless, the other 17 respondents were able to read the words clearly. Thus, the original type of font and size were retained.

A strength of the TBRSS is that it only needs the patient’s sociodemographic and medical history; it does not require any invasive or expensive investigation before its use as a screening tool. The patients can be assured that screening using the TBRSS will not cause any pain, therefore increasing the willingness of the patients to be screened using the TBRSS for TB disease. The users can also perform the screening process anywhere because it does not require any specific tools.

The TBRSS requires the user to calculate the total score to determine the patient’s risk of TB disease. This score may increase the patient’s awareness of their risk for TB disease and may lead to changes in behaviour. Previously, a patient was considered high risk if they were an active smoker. Our findings may help change patients’ perceptions of the risk of TB disease. They may be more willing to undergo a diagnostic test when they know that they have a high score on the TBRSS.

However, some limitations were also encountered in developing and validating the TBRSS. There are few variables that were reported to be associated with smoking and TB development, such as HIV status [40], weight maintenance [41] and occupational hazards (stone processing and silicosis) [42], which were not included as the study predictors. Response bias may have occurred when the team member interviewed the patients via telephone call or by face-to-face interview. Patients may give more desirable responses and prefer to hide the truth to avoid judgements of their lifestyle. To prevent this, the team members checked the information obtained from the interviews by reviewing the patients’ records in the health clinic and the TBIS101A records.

Some of the terms used on the TBRSS may not be familiar or well known to the user. Before the user can use the TBRSS, training or explanation may be required for them to understand the risk score fully, and not misinterpret the terms. Therefore, a list of operational definitions was also included in the risk score booklet to guide the users.

## 5. Conclusions

In this study, we were able to show with evidence that the TBRSS is a validated screening tool to be used to screen the smoker population. By involving significant predictors of TB when assessing the smoker population, we were able to apply and improve the existing conventional risk assessment tools that may help them to know their risk and come forward for TB diagnosis. If TB disease among smokers can be detected earlier, prompt treatment may help heal the smokers from the disease faster and reduce the risk of developing the complications of TB disease. Its use may help healthcare workers to conduct an effective screening process. Additionally, the newly developed risk score may help increase the detection of TB disease in general.

## Figures and Tables

**Figure 1 ijerph-19-06959-f001:**
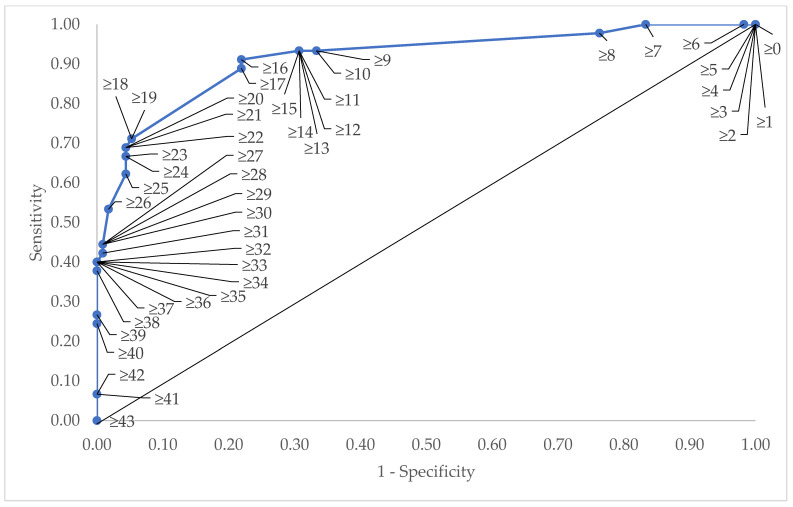
The sensitivity–specificity curve to determine the cut-off point for TBRSS.

**Table 1 ijerph-19-06959-t001:** Clinical characteristics of participants in Bachok, Kelantan (n = 159).

Variable	Cases (TB Disease) (n = 45)		Controls (None TB Disease) (n = 114)	
	n	(%)	n	(%)
**SOCIAL HISTORY**				
Age (years) mean (SD)	48.5	(17.)8	44.2	(15.3)
Gender				
Male	32	(71.1)	62	(54.4)
Female	13	(28.9)	52	(45.6)
Live in a TB area				
High incidence	10	(22.2)	10	(8.8)
Low incidence	35	(77.8)	104	(91.2)
Education status				
Primary school level	11	(24.4)	16	(14.1)
Secondary school level	29	(64.4)	60	(52.6)
Tertiary level/College/University	5	(11.2)	38	(33.3)
MEDICAL/SMOKING HISTORY				
Diabetes mellitus				
Yes	14	(31.1)	30	(26.3)
No	31	(68.9)	84	(73.7)
HIV status				
Yes	1	(2.2)	1	(0.9)
No	44	(97.8)	113	(99.1)
Immunosuppressed disease				
Yes	9	(20.0)	13	(11.4)
No	36	(80.0)	101	(88.6)
Smoking Status				
Active smoker	15	(33.3)	41	(36.0)
Ex-smoker	17	(37.8)	17	(14.9)
Passive smoker	13	(28.9)	56	(49.1)
Duration of exposure to smoke (years) mean (SD)	23.9	16.5	18.4	12.8
**TB HISTORY**				
Immunized with BCG	45	(100.0)	103	(90.3)
Expose to TB index case	18	(40.0)	35	(30.7)
Previous history of TB disease	6	(13.3)	4	(3.5)
**SYMPTOM HISTORY**				
Cough	36	(80.0)	101	(88.6)
Sputum	24	(53.3)	78	(68.4)
Weight loss	34	(75.6)	17	(14.9)
Having night sweat	22	(48.9)	1	(0.9)
Had chest pain	10	(22.2)	3	(2.6)
Loss of appetite	29	(64.4)	29	(25.4)
Fever	27	(60.0)	57	(50.0)
Hemoptysis	9	(20.0)	12	(10.5)
Symptom duation (days) *median (IQR)*	30	(76)	7	(11)

**Table 2 ijerph-19-06959-t002:** The associated factors with TB disease among smokers in Bachok, Kelantan (n = 159).

Variable	Crude OR	(95% CI)	*p*-Value ^a^
Live in a TB area			
High incidence	2.97	(1.13, 7.83)	0.026
Low incidence	Reference		
Education status			
Primary school level	5.23	(1.63, 18.94)	0.007
Secondary school level	3.67	(1.41, 11.53)	0.014
Tertiary level/College/University	Reference		
Smoking status			
Active smoker	1.58	(0.68, 3.71)	0.291
Ex-smoker	4.31	(1.77, 10.87)	0.002
Passive smoker	Reference		
Duration of exposure to smoke (years)	1.03	(1.01, 1.05)	0.030
Previous history of TB disease			
Yes	4.23	(1.15, 17.30)	0.032
No	Reference		
Cough			
Yes	Reference		0.162
No	1.94	(0.74, 4.90)	
Weight loss			
Yes	17.64	(7.76, 43.12)	<0.001
No	Reference		
Having night sweats			
Yes	108.09	(21.08, 1984.47)	<0.001
No	Reference		
Had chest pain			
Yes	10.57	(3.04, 49.14)	0.001
No	Reference		
Any loss of appetite			
Yes	5.31	(2.57, 11.38)	<0.001
No	Reference		
Duration of symptoms (days)	1.03	(1.01, 1.04)	<0.001

^a^ Simple logistic regression.

**Table 3 ijerph-19-06959-t003:** To determine the score for each of the variables.

Variables	*β*-Coefficient	Dividing with the Lowest *β*-Coefficient ^b^	Nearest Integer	Score Mark
Duration of exposure to smoke				
More than 10 years	0.459	1.723	2	3
Less than 10 years				1
Previous history of TB disease				
Yes	0.425	1.594	2	3
No				1
Presence of any cough				
Yes	0.267	1	1	2
No				1
Presence of significant night sweats				
Yes	4.615	17.319	15	15
No				1
Had significant weight loss				
Yes	2.590	9.717	10	10
No				1
Duration of any symptoms				
More than two weeks	1.762	6.611	7	10
Less or equal to two weeks				1

^b^ The lowest β coefficient is the β coefficient for cough, β = 0.267.

**Table 4 ijerph-19-06959-t004:** (**a**): The factor-content validation index (F-CVI) for each factor and scale-level content validity index, averaging method (S-CVI/Ave) of the respondents (n = 8). (**b**)The face validity index (FVI) for each factor (n = 20).

(**a**)		
**Factors**	**Total Score**	**F-CVI**
Smoking status	8	1.00
Duration of exposure to smoke	8	1.00
History of TB disease	8	1.00
Cough	7	0.88
Having night sweats	7	0.88
Had significant weight loss	8	1.00
Duration of symptoms	7	0.88
	**S-CVI/Ave**	**0.83**
(**b**)		
**Factors**	**Total Score**	**FVI**
Understanding of instructions given	20	1.00
Understanding of the sentences	20	1.00
Easiness to mark the score	20	1.00
Type and size of the font used	17	0.85
Appropriateness of the arrangement	20	1.00

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
