# Peer review of "Development and Validation of the Tuberculosis Risk Score for Smokers (TBRSS)"

_ijerph, 2022, doi:10.3390/ijerph19126959_

Round 1

Reviewer 1 Report

Comments to Authors 

    Tuberculosis (TB) remains a worldwide public health problem and Tobacco use, has long been under suspicion for associations with the TB epidemic. However, the links between the global tobacco and TB epidemics have not been recognized by public health managers until very recently.

    Tuberculosis is one of the leading infectious diseases for people living with HIV [1]. Baseline CD4 cell count, female patients, non-opportunistic diseases, and non-smoking status were negatively associated with the development of TB, whereas age of patients, living without partners, patients with no education, patients with low adherence, bedridden and ambulatory patients were positively associated to the development of TB in HIV patients [1]. 

    The high proportion of loss to follow-up among TB patients who smoke highlight the importance of providing early risk detection that examines the three main domains of risk factors such as socioeconomic, disease profiles and comorbidities [2]. Another, smoking is a risk factor for TB and weight maintenance (neither gaining or losing) after quitting smoking might reduce the risk of TB development [3].

    In addition, false-negative Interferon-Gamma Release Assays' obtained on active TB patients exposed to second-hand smoke, together with the decrease of specific cytokines released, suggest that tobacco may alter the immune response [4].

    In conclusion, prevention and protection rules are needed to be enforced in the occupation involving artificial stone processing; smoking may be associated with declined lung function in artificial stone patients [5].

    I think that this findings too emphasise the importance of the development and validation of the Tuberculosis Risk Score for Smokers.

References

  1. Risk Factors for the Development of Tuberculosis Among HIV-Positive Adults Under Highly Active Antiretroviral Therapy at Government Hospitals in Amhara Region, Ethiopia. Int J Gen Med. 2022;15:3031-3041. Published 2022 Mar 15. doi:10.2147/IJGM.S358517.
  2. Characteristics and determinants of loss to follow-up among tuberculosis (TB) patients who smoke in an industrial state of Malaysia: a registry-based study of the years 2013-2017. BMC Public Health. 2022; 22 (1): 638. Published 2022 Apr 1. doi:10.1186/s12889-022-13020-3.
  3. Association of weight change following smoking cessation with the risk of tuberculosis development: A nationwide population-based cohort study. PLoS One. 2022; 17 (4): e0266262. Published 2022 Apr 7. doi:10.1371/journal.pone.0266262.
  4. Tobacco Smoking and Second-Hand Smoke Exposure Impact on Tuberculosis in Children. J Clin Med. 2022; 11 (7): 2000. Published 2022 Apr 2. doi:10.3390/jcm11072000.
  5. Risk Factors of Silicosis Progression: A Retrospective Cohort Study in China. Front Med (Lausanne). 2022; 9: 832052. Published 2022 Apr 4. doi:10.3389/fmed.2022.832052.

Reviewer 2 Report

My major concern is the reason to develop a tool for screening TB in smokers. Although smokers may have a higher risk of contracting TB compared with non-smokers, the content of Introduction section in this present manuscript did not support the necessity to develop a tool for screening TB in smokers. Are the screen methods for TB in the general population not fit for smokers? Does the new tool developed in this study have a higher adequacy to identify the smokers with TB than the tools used in the general population? Is there any situation in which only smokers are the candidates for screening and have to be screened by a specific tool such as the tool developed in this study? Why smokers can not be screened by the symptoms of TB? Is it a more practical method to encourage smokers to receive a x ray examination? Before answering these question, the necessity of developing a screening tool for TB in smokers is questionable.

Reviewer 3 Report

Abstract: The Study was developed in a very short period, about six months, would it be possible to consider it as a preliminary phase?

Keywords: add "TBRSS, TB OUTCOME"

Introduction:

Line 32 insert reference(s).  -

What is the epidemiology of TB in Malaysia?  -

From line 47 to 51, I think it is better to transfer it at the beginning of Materials and Methods. -

In line 47 u wrote "active smoker", What do you mean with it? Have you established a minimum number of cigarettes to include patients in the study? -

Line 65-66 the timing of the study is different from the one presented in the abstract.

Materials and Methods: 

TBIS101A / TBIS102A explain what they are. -

Line 74 - 75 are in contradiction with line 47 "active smoker"

Results:

Line 140 write in full mode the acronimous BCG

Line 159-160 If these four variables still  have an impact, even if minor, they cannot be included as minor criteria?

Increase the conclusions

References: the references present a few recent articles. I think it could be useful adding two articles in introductions:

- Napolitani M., Kundisova, L., Messina G., Nante N.; European J. Public Health (Marseille, 20-23 Novembre 2019), 29, 4, 330, 2019. The applicability of methods for assessing comorbidities: The Cumulative Illness Rating Scale (CIRS)

Napolitani M., Guarducci G., Albinova G., messina G., Nante N.; Int. J. Res. Public Health, 19, 3452, 1-14, 2022; How to improve the drafting of health profiles

Round 2

Reviewer 2 Report

I still have the concern about the necessity of developing a new screening tool for smokers with TB in community. The authors did not provide the answer for why the currently used tools for screening TB among smokers visiting the treatment units can not be applied to smokers in community.
